# Exploring the knowledge, attitudes, and practice towards child eye health: A qualitative analysis of parent experience focus groups

Sadik Taju Sherief [1,2]*, Samson Tesfaye[3], Zelalem Eshetu[4], Asim Ali[5], Helen Dimaras[2,5]

1 Department of Ophthalmology, Addis Ababa University, Addis Ababa, Ethiopia, 2 Child Health Evaluative Sciences Program and Centre for Global Child Health, Sickkids Research Institute, Toronto, Canada, 3 Orbis International Ethiopia, Addis Ababa, Ethiopia, 4 Biruh Vision Speciality Eye Center, Addis Ababa, Ethiopia, 5 Department of Ophthalmology and Vision Sciences, The Hospital for Sick Children and University of Toronto, Toronto, Canada

* goge4000@yahoo.com

**Data Availability Statement:** All relevant data are within the paper and its Supporting Information files.

## Abstract

### Background

The majority of childhood blindness causes in low-income countries are treatable or avoidable. Parents or guardians are responsible for making decisions regarding a child's eye care. Understanding parents' awareness and perception of eye problems in crucial in helping to know parents' eye care-seeking behavior.

### Objective

To determine parental knowledge, attitudes and practice regarding child eye health.

### Methods

Seven focus groups were carried out in Northwest Ethiopia on knowledge, attitude and practice of parents towards child eye health. Their responses were tape-recorded and later transcribed. A thematic phenomenological approach was used for the analysis.

### Result

Seventy-one parents participated in the focus groups. Participants were aware of common eye problems like trachoma, trauma, and glaucoma. However, they were unaware of the causes and etiologies of childhood blindness. Participants perceived that eye problems could be treated with hygiene and food, and often held misconceptions about the cause of strabismus and utilization of wearing spectacles.

### Conclusion

The study revealed that parents are often unaware of the causes and etiologies of common childhood eye diseases, which has downstream effects on health-seeking behavior. Health promotion efforts, potentially through mass and social media, could be helpful to raise

**Funding:** Light for the World financially supported the field work of this study. The funders had no role in study design, data collection and analysis, decision to publish, or preparation of the manuscript.

**Competing interests:** The authors have declared that no competing interests exist.

awareness, coupled with training of health professionals at primary and secondary health facility levels.

## Introduction

Control of blindness in children is closely linked to child survival [1]. Without initiatives to stop preventable childhood blindness, it is anticipated that by 2020 the world would have lost more than US$110 billion [2]. In low-and-middle income countries, up to 60% of children die within a year of going blind–usually due to underlying conditions such as measles, meningitis, rubella, prematurity, genetic diseases and head injuries [3,4]. In 2020, the estimated percentage of Gross Domestic Product (GDP) loss due to blindness in sub-Saharan Africa was 0.5% [5].

The prevalence of childhood blindness is closely linked to the socioeconomic development of the country and the mortality rate of those under five years of age [1,6]. An estimated 19 million children are visually impaired globally, of which 1.26 million children are blind [7,8]. Almost 3/4 of these children live in low-and-middle income countries (LMICs) [9], where there are limited data on prevalence and causes of childhood blindness. In sub-Saharan African countries the prevalence rate of child blindness is to be 1.5 per 1000 children, based on an extrapolation model using under five mortality rates [1,9].

There are several dimensions affecting child eye health (CEH) status, including the availability and quality of eye health care professionals, accessibility of eye care (i.e. affordability, location) and public awareness of early signs and symptoms of eye disease [1]. The high prevalence of childhood visual impairment in sub-Saharan Africa has been attributed to lack of availability, accessibility, and affordability of eye care services [10]. Understanding barriers and facilitators of eye care service utilization is key to addressing visual impairment in Africa [11]. Common barriers to seeking eye examination for children include: limited finances, long distance between residence and the health facilities and the broader availability and lower cost of traditional medicine [12–14].

Early detection of childhood eye diseases at the household and community levels is essential in reducing the global burden of visual impairment and childhood blindness. In particular, parental knowledge, attitude and practice toward CEH plays an important role in early detection and prompt care; when parents are able to recognize vision issues in children and seek the necessary eye care, burden of disease is dramatically improved [15]. Parental knowledge childhood eye disease has been shown to be a predictor of seeking eye care for children [16].

However, poor parental awareness of common eye diseases and their prevention is one of the main challenges in the fight against childhood blindness [17–19]. Figures from India [20], Ethiopia [21] and Ghana [12,22] indicate the proportion of parents with good knowledge of childhood blindness ranges from 9% to 76%. Poor knowledge is associated with lower levels of education [22]. In Malawi, low parental literacy was associated with lower likelihood of parental consent to eye surgery for children [23]. Poor knowledge can lead to neglect of childhood eye problems or attempt to self-medicate or treat with traditional methods [24]. In some parts of Africa, widely held misconceptions prevail due to unique cultural and social factors; for example, congenital blindness is sometimes thought to be incurable, and caused by ancestral curse [2].

In Ethiopia, the prevalence of childhood blindness is 0.1% and accounts for over 6% of the total blindness burden [25]. With the total population projected to reach 144,944,000 by 2030 and double (205,411,000) by 2050 [26], prevention of childhood blindness becomes a pressing concern. Northwest Ethiopia in particular is more vulnerable to the burden of childhood

blindness, as eye care facilities are limited. Furthermore, the distinct culture and societal norms in this region necessitates a closer look at the awareness, perceptions and beliefs of parents towards CEH and their potential influence on childhood blindness. Qualitative studies in particular, have the potential to explain the nature and kind of barriers in CEH, yet they are rare in the pediatric ophthalmology literature [27]. The aim of this study, therefore, was to perform a qualitative study to assess knowledge, attitudes and practice among parents towards CEH in Northwest Ethiopia.

## Methods and materials

### Study design and setting

A cross-sectional, qualitative study using focus group discussions (FGD) was conducted to obtain in depth insights of the knowledge, attitude and practice of parents towards CEH. The study was carried out in June 2022 in the North Gondar Zone of the Amhara region of Ethiopia, located 745 km northwest of Addis Ababa. The Zone has a total population of 3,999,113 with about a 1:1 male to female ratio [28]. Metema, Debark, and University of Gondar Hospital are three hospitals located within the Zone; the University of Gondar Hospital is the only tertiary eye care facility in the Zone.

### Study participants and recruitment

The participants included parents having children age less than 12 years of age. Participants were drawn from two Woredas (i.e., third level administrative districts) in the Central Zone of the Amhara region. To recruit a diverse set of study participants, study investigators sent an introductory study letter to each Woreda office asking for their support to help identify suitable candidates. Participants were purposively selected to ensure diverse representation of demographic factors and employment type (e.g. health extension workers (HEW), farmers, teachers, religious or community leaders) where possible.

### Data collection

FGD were conducted by two local health systems researchers, who received 2 days of training in qualitative data collection for the purpose of this study. Semi-structured discussion guides were developed in English, translated into Amharic language and back-translated for consistency. The discussion guides were piloted in a kebeles (i.e., smallest administrative districts in Ethiopia, akin to a neighborhood) that was not included in the study, and adaptations to questions were made. Demographic information (e.g. sex, age, kebele, religion, literacy, employment status, employment type) was collected for each participant prior to beginning the FGD. The FGD interview guide included questions on (i) knowledge (e.g. information about child eye care, knowledge about poor eyesight, barriers for accessing child eye care}, (ii) attitudes (e.g. towards spectacle wear) and (iii) practice (e.g. prevention of visual impairment, treatment and impact of visual impairment). Daily debriefing meetings with all data collectors were organized to examine major findings, review saturation, and refine research directions. All FGDs were tape recorded, transcribed and translated into English. The discussions lasted about 60 to 90 minutes and included 7–13 participants each. Recruitment of participants ended when data saturation was reached.

### Data analysis

The transcripts were read to identify key themes and develop a coding framework. To do this, we first loaded the translated data into Open Code 4.03 (Umea University, Sweden), an open

source computer software for analyzing qualitative data. A thematic phenomenological approach was employed to analyze the data. The coded transcripts were further analyzed and summarized into narratives for each theme and sub-theme.

## Results

### Study participants

Seventy-one parents of mean age 41.1(±11.1) years participated in 7 FGD (S1 Table). Among the participants, 55 (77%) were males and 16 (23%) were females. The mean age of the participants was 41.1± 1.1 (range 26–69) years and 30 (42%) of them were in the age group of 30–39 years. In terms of employment, 19 (27%) were farmers, 14 (20%) were teachers, and 7 (12.7%) were HEWs. The majority of participants (53/71, 75%) had at least finished high school. The majority of participants (61/71, 86%) were Orthodox Christians (Table 1).

### Focus group discussions

Seven major themes emerged from the thematic analysis. Two themes were related to Knowledge: (1) Information on CEH; and (2) Knowledge about child eye problems. Three themes were related to Attitudes: (3) Barriers to child eye care; (4) Use of eye glasses; and (5) Impact of childhood blindness. Two themes were related to Practice: (6) Treatment and prevention of CEH problems and (7) Cultural and traditional practices. S2 Table describes these themes, sub-themes and quotes identified during analysis, which are further discussed below.

**1.0 Information on CEH.** *1.1. Parents accessed child eye health (CEH) information from media, health professionals and patients.* Most participants reported obtaining information on CEH from health professionals (during their visits to health facilities or during home-to-home visits) or the media, such as radio (mainly in the form of advertisements about eye health). For some of the study participants, sick persons were considered a source of health information. One participant emphasized that the advantage this approach was that:

> *". . .getting an information from a patient is good because we can have complete information from that person, including prevention means and where we can get treatment if we need it."– Local leader.*

*1.2. Content of CEH information was related to hygiene, optimal health visits, and avoiding bright lights and television.* Participants cited that they received instruction on: personal and environmental hygiene (particularly to washing hands frequently and keeping eyes clean); when to go to health facilities; avoiding bright light; and minimizing television time. However, almost all participants emphasized that they felt that information they had access to about CEH was inadequate.

**2.0 Knowledge about child eye problem.** *2.1. Cause of eye problems were many and varied.* The participants outlined trachoma, glaucoma, exposure to dirt particles, and trauma to be the causes of poor eyesight in children. Misaligned eyes (strabismus) was another condition that participants mentioned as commonly observed in their communities. Regarding the cause of strabismus almost all of them mentioned "staring at light" as the cause. One participant said;

> *"Traditionally our mothers would say when children see into the dark their eyes tend to deviate. This can be avoided by preventing the children's view by blocking cloth or a curtain so they wouldn't see towards bright lights at night." -Priest.*

**Table 1. Characteristics of FGD participants.**

| Variable | | n | % | mean (± SD) | range |
|---|---|---|---|---|---|
| **Total Study Participants** | | **71** | **100%** | | |
| **Sex** | | | | | |
| | Male | 55 | 77% | | |
| | Female | 16 | 23% | | |
| **Age (years)** | | | | 41.1(±11.1) | 26–69 |
| | 20–29 | 8 | 11% | | |
| | 30–39 | 30 | 42% | | |
| | 40–49 | 18 | 25% | | |
| | 50–59 | 7 | 10% | | |
| | 60–69 | 8 | 11% | | |
| **Kebele(Neighbourhood)** | | | | | |
| | Aba Jale Kebele 11 | 13 | 18% | | |
| | Adebabay Iyesus 04 | 12 | 17% | | |
| | Shwa Ber Kebele 18 | 12 | 17% | | |
| | Azezo Kebele 20 | 9 | 13% | | |
| | Arbaya | 7 | 10% | | |
| | Worehala | 9 | 13% | | |
| | Kalay | 9 | 13% | | |
| **Education** | | | | | |
| | Illiterate | 5 | 7% | | |
| | Read and write | 6 | 8% | | |
| | Elementary school | 7 | 10% | | |
| | High School | 26 | 37% | | |
| | Diploma and above | 27 | 38% | | |
| **Religion** | | | | | |
| | Orthodox Christian | 61 | 86% | | |
| | Muslim | 10 | 14% | | |
| **Employment Status** | | | 0% | | |
| | Employed | 68 | 96% | | |
| | Farmer | 19 | 27% | | |
| | Teacher | 14 | 20% | | |
| | Local Leader | 8 | 11% | | |
| | Government Employee | 7 | 10% | | |
| | Health Extension Workers | 7 | 10% | | |
| | Priest | 7 | 10% | | |
| | Sheikh | 5 | 7% | | |
| | Leader of Youth Association | 1 | 1% | | |
| | Unemployed | 3 | 4% | | |
| | Housewife | 2 | 3% | | |

*2.2. Signs of vision impairment in children included falling, difficulty reading from afar or seeing at night, and poor school performance.* The participants believed that children with poor eyesight would have difficulty seeing at night and in reading. They described the need to sit at the front of the classroom or near the television to see more clearly. However, almost all participants didn't know when they would seek care for such issues. One of the parent participants, who also worked as a HEW, said:

*"I personally have found many children who had squint, but I had no idea if it was treatable. And hence, I have not told them to seek treatment. But I know if they have decreased vision, it can be treated at hospital".* -HEW

Regarding seeking child eye care, the majority of the study participants mentioned "pain and redness" as a reason for seeking treatment. Another participant described his take as:

*"Eye treatment needs high level of care, but we have little knowledge and we do not seek medical care early. When our visual ability decreases, we have difficulties to read distant writings. We need to take a closer look to read. Especially when we have tear or burning sensations to our eyes, we need to seek eye treatment".* -Teacher.

**3.0 Barriers to child eye care.** *3.1 Financial constraints, distance and poor knowledge impede access to CEH services.* Most participants put financial issues (inability to afford treatment) and lack of awareness as forerunners that affect access to child eye care.

Some also raised the issues of lack of transportation, less emphasis by government on delivery of CEH services, and limited media attention to CEH issues, as barriers to access CEH service. One participant stated his dismay as follows:

*"Poor attention is given to children eye care. Starting from clinics and hospitals including health minister give poor attention to children eye care and I believe this is one obstacle. In seeing the society there could also be limitation in the awareness".*—Leader of Youth Association.

*3.2 Long wait times and substandard quality of health institutions impede provision of CEH services.* Several participants raised health provider-related issues as barriers to accessing CEH services. These included absence of the correct health professional (e.g. eye specialist), very long wait times, and referral to facilities far from their home (usually to the University of Gondar). One participant, described his concern about health provider quality as follows:

*"I am sure you know what a pig is. A pig comes out of the woods intentionally to wash itself, but after it's done washing it rolls in the mud. If the medias are supported financially and they present the eye health need well, but if the service at the health centers is substandard it will be like the pig story".* -Teacher

**4.0 Use of eye glasses.** *4.1 Eye glasses perceived to cause harm and stigmatize children.* Some of the participants had concerns about the use of eyeglasses for children, as they had limited experience with children wearing spectacles. One participant discussed his concern as follows:

*"Till now I have never seen or heard anybody teaching children to use eyeglasses. And I believe it will be better if the health care givers get a lot involved in creating awareness".* -Teacher.

Some of them even believe that vision of children wearing spectacle will deteriorate. And another participant said:

*"People advise against routine use of eyeglasses because they believe it decreases vision. If the eyeglass is given by a professional after thorough examination, they will use it. The main thing is, I believe that attention should be given to other treatments and glasses be made the last*

*resort and that other treatments are given to maintain their eyesight, keep their hygiene, but the main thing is that the health professional creates awareness at the lower levels including woreda and kebeles." -Sheikh*

Another participant put his opinion towards children wearing spectacles as:

*"My personal attitude towards using eyeglasses in children is, if someone in my family is pre-scribed eyeglasses, I am not willing because it will be a bad thing if they start to wear glasses from this age. If they start to wear glasses at a young age, they could be short sighted". -Farmer.*

*4.2 Eye glasses perceived to improve vision.* Some participants perceived wearing spectacles as useful, as they improve vision (e.g., important for reading), protect eyes from dirt and bright lights and protect from illness. However, the majority of the participants mentioned stigma against children wearing spectacles as a concern.
One participant said:

*"I believe wearing glasses has benefits and no harm. In kids usually wearing glasses is seen as being lame and being outcasted by the other students and this is due to lack of awareness". -Teacher.*

**5.0 Impact of childhood blindness.** *5.1 Long term consequences include poor school performance and failure to progress in life.* The impact of visual impairment on children was outlined by the participants as poor school performance and repeatedly failing to progress. They also said children would sustain psychological trauma as the result of insults from their peers and teachers and being labeled as "stupid".
One of the school teachers described the following regarding the impact of poor vision in education as:

*"Visual problems by themselves are one of the reasons for poor school performance. One of my students had terrible handwriting and struggled to keep up with the blackboard. Her results were dismal, and she failed that academic year. She keeps asking to be in the front chair, she mentioned she can't see from a distance. When eye doctors came to our school for a free eye exam, I took her for an eye exam. They also informed us that she requires glasses. Sadly, her poor performance was due to her eye problem.".–Teacher*

**6.0 Prevention of childhood blindness.** *6.1 Prevention of vision impairment includes having good personal hygiene, eating a healthy diet and avoiding sunlight.* The participants stated the importance of keeping good personal and environmental hygiene and eating a balanced diet (e.g. eating carrots was mentioned specifically) for preventing visual impairment in children. They also indicated that eye health education and messaging was important for promoting the use of these practices.
When it came to seeking care, none of the participants had taken their children to receive an eye examination. Most participants said they would seek treatment specifically if their children's vision was reduced. One participant said:

*"To prevent reduction of vision the major thing is maintaining their hygiene. The other thing is going to health centers. We can't say that eye care is fully available in the society. For example, eye care is given at a hospital level in only one institution and so that the children of the community have access to this, there should be some reforms."-Teacher*

And another participant expressed his position as follows:

*"I tell my children not to read in sunlight and try to keep their personal hygiene Promoting the habit of using eyeglasses, avoiding reading in sunlight and keeping hygiene can prevent eye diseases."* -Government employee.

**7.0 Cultural practices & traditional medicine.**   *7.1 Cultural terminology & practices.* Participants spoke about childhood diseases that have specific cultural names, perceived causes and traditional treatment. "Alemochie" is a cultural nomenclature for measles and traditionally it has its own treatment, described as follows by one participant:

*"Alemochie' means a word to describe when children develop fever, body rash, and red eyes, they mix incense, niger, and roasted barley, after which they will climb a tree and a traditional custom starts."*–woman Local leader.

*7.2 Traditional medicine has a role in CEH.* For some of the study participants, traditional medicine was easily accessible and affordable. One participant described his observation on the cost of eye care and waiting time at health facilities as follows:

*"Some people seek for traditional treatment because they cannot afford to go to health facilities. And even if they go, they spend hours in the waiting room waiting for their turn before they get the treatment. So, I think it will be better if there is a separate institution for treating these conditions."* -HEW

Primary health care workers participated in this study mentioned that vising traditional healers is a common practice among parents. One of the HEW said:

*"We tell the mothers to take the children to the hospital to be evaluated, but some go towards traditional medicines."* -HEW

## Discussion

This qualitative study set in Northwest Ethiopia revealed gaps in parental knowledge towards child eye health, as well as attitudes and practices that may interfere with prevention and treatment of visual impairment and blindness in children. The study findings suggest 5 key areas for action to improve knowledge and health-seeking behaviour of parents, discussed in the sections that follow.

### Action #1. Improve CEH knowledge among parents via trusted sources

Our study participants indicated that their primary sources of CEH information were the media and health professionals, consistent with prior studies on parental access to health information [29,30]. However, participants acknowledged that they lacked sufficient knowledge regarding CEH, and our results demonstrated this as they misunderstood the etiologies and common causes of child eye blindness in their setting. Availability of information does not equate to knowledge creation, and it is well acknowledged that continuous knowledge translation strategies are required to educate the lay public on health issues, particularly in low resource settings, which have unique geographical, financial, and cultural considerations [31]. In order to overcome these conventional hurdles to access health information, some have argued in favor of using digital health services to distribute health information [32–34]. The

increase in the internet penetration rate from 7.5 to 25% in the past 10 years in Ethiopia gives an opportunity to use the internet as a source of health information in the coming years [35]. However, such activities should be coupled with campaigns to educate the public on trustworthy sources of information, such as the Ministry of Health or other relevant professional bodies, such that misinformation is avoided. People with lived experience of vision impairment or blindness could be another knowledgeable and trusted source of information, as noted by a participant in our study. Health champions have been instrumental in raising awareness and reducing stigma for HIV/AIDS [36], and could have an equivalent role for CEH awareness in Ethiopia.

## Action #2. Directly Address Common Misconceptions and Myths about Causes of Blindness and Use of Eye Glasses

An estimated 100,000 blind youngsters reside in Ethiopia alone. According to a 2017 study at a school for the blind in the same region as the current study, corneal scarring from vitamin A deficiency and measles are the leading causes of blindness [37]. Yet these were not noted by the participants in our study, who instead identified trachoma, glaucoma, and trauma as the primary causes of childhood blindness. This difference warrants further investigation, since in LMIC most research on childhood blindness is performed in schools for the blind, yet the conclusions drawn may not be representative of the community, potentially limiting the generalizability of the findings.

Moreover, no participant mentioned amblyopia as a childhood vision concern. This finding is consistent with research from India [19] and Nigeria [38] where in each study, only one parent was aware of amblyopia. In contrast, a research study from Saudi Arabia revealed that parents' understanding of amblyopia was the greatest among other disorders [39]. Amblyopia can be prevented and treated, particularly if it is detected early in children under the age of eight [40], thus parental knowledge of this is essential.

Our study also revealed a myth in the community related to strabismus causation, where parents believed that strabismus was caused by staring at light. A prior cross-sectional community survey in Southern Ethiopia revealed that many of the participants had false beliefs regarding the cause of strabismus [41]. Similarly, a study carried out in Nigeria demonstrated that a common misconception was that strabismus could result "if a baby's sleep is not changed from time to time" [42]. A study from Chennai, India indicated that the majority of parents in that setting believed that strabismus was a sign of good fortune and did not seek care for [19]. Sometimes, strabismus can be the first sign of ocular tumor, like retinoblastoma [43]. Poor parental education, misunderstandings, and misinformation have a negative impact on the age of strabismus presentation and management, which differs from country to country [44].

This study also revealed differences in perceptions among the study participants with regard to the use of eye glasses in children. Several participants held the misconception that wearing eyeglasses on a regular basis could impair vision. Furthermore, the stigma associated with wearing glasses was a major worry raised by the majority of participants. Similarly, study from India showed that parents were hesitant to provide spectacle correction for their children with vision problems because they perceived wearing spectacles as a stigma [19]. One study found that parents believed no child under the age of four should wear glasses at any time [45]. In a Nigerian study, parents commonly believed that children could become overly reliant on eye glasses and be unable to function without them [38]. Parental disapproval has been reported as one of the main reasons for the non-compliance with spectacle utilization among school students [46]. Identifying reasons for non-compliance with spectacle use is important

for understanding the social determinants of intervention. This will help to enhance the awareness of the parents of children with refractive error, by informing health education programs.

### Action #3. Educate Parents on When to Seek Eye Care for Children

Despite the above-mentioned misconceptions and gaps in knowledge, participants in this study were able to recognize signs of poor vision in children. They also were aware of negative effects of poor vision, noting how children with visual impairment tend to do poorly in school and repeatedly fail to progress. Similar findings have been shown in studies from Eswatini, Nigeria, Saudi Arabia and India [19,38,39]. Yet, our participants were unable to pinpoint the right moment to seek medical assistance, and none of the participants reported taking their children to the eye doctor for a routine checkup. Poor parental health-seeking behavior has been attributed to a lack of health information [17,47]. A European study of parental health-seeking behavior indicates that upon first notice of signs and symptoms parents initially attempt to manage the condition at home, and tend to seek medical attention when they feel they have "lost control of the situation" [48]. Improving the messaging to parents on when to seek care for routine visits as well as upon notice of CEH issues is essential to prevention of childhood blindness.

### Action #4. Improve Capacity and Quality of CEH Services

Participants noted several barriers in accessing services in the context of Northwest Ethiopia. These included financial constraints, a lack of public knowledge, a shortage of qualified eye health specialists, lengthy wait times at tertiary eye care facilities, and a disregard for CEH by the health system. This is consistent with the literature showing that non-availability, non-accessibility, and non-affordability of eye care are three primary factors that contribute to the high prevalence of childhood visual impairment in Africa [2,10,11,13,29,38,49–55]. Furthermore, ineffective policies and a lack of facilities have also been mentioned as a major barrier for child eye health, particularly in low-income nations [2]. Our study highlighted that parents' health-seeking behavior was influenced by perceived substandard health facilities. Numerous studies have revealed that socioeconomic position, access to healthcare facilities, and perceived service quality are important determinants of maternal and child health seeking behavior among different population groups [56–58]. Therefore, improving the health seeking behavior of parents must go hand in hand with improving capacity to provide excellent child eye care.

### Action #5. Establish a Collaboration Between Traditional Healers and CEH Services

Traditional medicine was identified by study participants as one of the main treatment alternatives for CEH. While not explicitly stated by participants, traditional medicine presumably evades the access challenges of cost and distance, and is likely aligned with deep-rooted cultural beliefs and terminologies revealed in our study. Studies from Ethiopia showed that nearly 80% of the population relies heavily on herbal treatments as their primary source of therapy, and traditional eye medicine is widely used in the country [59–61]. However, there is no formal connection or collaboration between traditional medicine practitioners and the Ethiopian health system. A study from Eswatini showed that parents believed that traditional healers could play a role in disseminating eye health information [29]. Research conducted in Zimbabwe and Malawi suggests that a sustainable collaborative program between traditional and modern practitioners could help to reduce preventable blindness by advising against the use of harmful traditional practices [62]. Given the high use of traditional medicine for eye care, such collaborations may have utility in Ethiopia as well.

## Study limitations

A main limitation was the gender imbalance in the study participants, with more fathers taking part in the study. While in many parts of Africa, females are generally the ones who take care of children at home, in Ethiopia, fathers play an important role in caregiving and are often the ones who accompany children to health visits [63]; this may have been a reason for the gender imbalance in the study.

## Conclusion

In conclusion, this study captured the current state of knowledge, attitude and practice towards child eye health among parents in Northwest Ethiopia. The study revealed concerning gaps in knowledge and incorrect views on strabismus, spectacle utilization, blindness prevention and the etiologic causes of the most common childhood eye diseases. Traditional medicine and practices were perceived to have a role in CEH. Study participants revealed a desire for more health information on child eye diseases from trusted sources. Collectively, study results suggest five action items to improve knowledge and health-seeking behaviour of parents: (1) improve CEH knowledge among parents via trusted sources; (2) directly address common misconceptions and myths about causes of blindness and use of eye glasses; (3) educate parents on when to seek eye care for children; and (4) improve capacity and quality of CEH services; and (5) establish a collaboration between traditional healers and CEH services. These actions may aid in reducing childhood blindness in Ethiopia.

## Supporting information

**S1 Table. Contains sex and age details of focus group discussion participants.**
(DOC)

**S2 Table. Contains themes, subthemes, and representative quotes.**
(DOC)

## Acknowledgments

Our special appreciation goes to Suadik Hassen, Alem Mekonen, Eyoel Lemma Semalign Abew and Demeke Debebe for their support and facilitation. Salient gratefulness is extended to all the study participants, community representatives and others who participated in the discussions by providing all the necessary information for the study.

## Author Contributions

**Conceptualization:** Sadik Taju Sherief, Zelalem Eshetu, Asim Ali, Helen Dimaras.

**Data curation:** Samson Tesfaye, Zelalem Eshetu, Helen Dimaras.

**Formal analysis:** Sadik Taju Sherief, Samson Tesfaye, Zelalem Eshetu, Asim Ali.

**Investigation:** Sadik Taju Sherief.

**Writing – original draft:** Sadik Taju Sherief, Samson Tesfaye, Zelalem Eshetu, Asim Ali, Helen Dimaras.

**Writing – review & editing:** Sadik Taju Sherief, Samson Tesfaye, Zelalem Eshetu, Asim Ali, Helen Dimaras.

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
