## [Decision Letter · Decision Letter 0]

10 Aug 2023

PONE-D-23-07767Exploring the knowledge, attitudes, and practice towards child eye health: A qualitative analysis of parent experience focus groupsPLOS ONE

Dear Dr. SHERIEF,

Thank you for submitting your manuscript to PLOS ONE. After careful consideration, we feel that it has merit but does not fully meet PLOS ONE’s publication criteria as it currently stands. Therefore, we invite you to submit a revised version of the manuscript that addresses the points raised during the review process.

We look forward to receiving your revised manuscript.

Kind regards,

Ugochukwu Anthony Eze

Academic Editor

PLOS ONE

Journal Requirements:

“We would like to thank Light for the World for providing financial support for this study”

“Light for the World financially supported the field work of this study.

The funders had no role in study design, data collection and analysis, decision to publish , or preparation of the manuscript.”

Additional Editor Comments (if provided):

Haven read through your work and the review comments. The following suggestions have been made as seen in the attached comments. If you want to continue publishing this work with the journal, you are advised to revise as suggested.

It is important to note that the references were not well done and generally inconsistent. Quite a number require review and include number 19, 24, 25, 27, 30, 31, 32, 33, 34, 48, 41, 43, 44, 45, 46, 47, 48, 52. Reference Number 32 and all internet source were all wrongly written. For the internet source date of retrieval of information are important.

Kindly acquaint yourselves with the style of the journal. Kindly also remove the months in the journal articles listed. The years are sufficient. Below are 2 example extracted from a PLos ONE publication.

Kindly pay attention for journal and website sources.

Kanmodi K, Ekundayo O, Adebayo O, Efuntoye O, Ogunsuji O, Ibiyo M et al. Challenges of Residency

Training and Early Career Doctors in Nigeria Study (CHARTING STUDY): A Protocol Paper. Niger J

Med. 2019; 28(2):198–205.

Nigerian Association of Resident Doctors. Available at https://en.wikipedia.org/wiki/Nigerian_

Association_of_Resident_DoctorsWikipedia. Accessed 14 July 2021.

Reviewers' comments:

Reviewer's Responses to Questions

**Comments to the Author**

1. Is the manuscript technically sound, and do the data support the conclusions?

Reviewer #1: Partly

Reviewer #2: Yes

Reviewer #3: Yes

Reviewer #4: Yes

2. Has the statistical analysis been performed appropriately and rigorously? 

Reviewer #1: N/A

Reviewer #2: Yes

Reviewer #3: Yes

Reviewer #4: Yes

3. Have the authors made all data underlying the findings in their manuscript fully available?

Reviewer #1: No

Reviewer #2: Yes

Reviewer #3: Yes

Reviewer #4: Yes

4. Is the manuscript presented in an intelligible fashion and written in standard English?

Reviewer #1: No

Reviewer #2: Yes

Reviewer #3: Yes

Reviewer #4: Yes

5. Review Comments to the Author

Reviewer #1: The study looked at exploring parental KAP on children’s vision.

1.The literature review could be more informative by focusing on the topic itself – parental KAP towards CEH. There is very little on this in the introduction.

2.What exactly is the novelty of this research? It seems like a lot has been known on this topic.

Alrasheed SH, Naidoo KS, Clarke-Farr PC. Childhood eye care services in South Darfur State of Sudan: Learner and parent perspectives. African Vision and Eye Health. 2016 Jan 1;75(1):1-3.

Malek Shahenaz M, Karelia Bharti N. Knowledge, Attitude And Practice On Myths And Belief Of Parents, Regarding Childhood Illness In Outdoor Unit Of Gynaecology Department Of A Tertiary Care Teaching Hospital.

Molla A, Meshesha B, Delelegn D, Reddy PS. Knowledge, attitudes and practice (KAP) and associated factors of parents/guardians on childhood eye care, Sidama Zone, Southern Ethiopia. Int J Sci Res. 2014;3:339–42.

3.To improve the state of CEH, there are plenty of factors – parental KAP is one of them. Can the authors describe these factors in their context, how they have/have not been dealt with and how parental KAP fits in the puzzle.

4.How was the sample size of 7 focus groups determined? No sampling frame was provided.

5.There is no such thing as no risk in research – perhaps minimal risks?

6.How was “Knowledge about child eye problems; attitude related themes” not fall under Attitude? A confusing way of presenting your findings.

7.It is uncommon to present findings both in table 2 and repeat them in the narrative. Choose one.

8.The subthemes could be more informative – e.g., instead of Source of Information, it could be “Parents accessed child eye health information from X,Y,Z.

9.I would be interested to look at the codebook seeing the dataset has not been shared.

10.“According to a 2017 study at a blind school in the same region as the current study, corneal scarring from vitamin A deficiency and measles are the leading causes of blindness [25]. Yet these were not noted by the participants in our study, who instead identified trachoma, glaucoma, and trauma as the primary causes of childhood blindness.” – perhaps your participants do not have children from the blind schools? Please substantiate your findings with relevant literature.

11.The primary sources of information, according to study participants, were the media and health professionals. However, they all acknowledged that they lack sufficient knowledge regarding children's eye health. This may be the cause of parents misunderstanding the etiologies and common causes of child eye blindness in their setting. – How do you relate misunderstanding the etiologies to sources of information being media and health professionals?

12.Overall, the Discussions section is very sporadic with multiple themes in one paragraph. The team could mainstream one point per paragraph.

13.You might need to consult a native English speaker to smoothen the academic writing and grammatical errors.

Reviewer #2: Th authors set out to explore the KAP of parents regarding their ocular health seeking behaviour on behalf of their children. The premise of the study is justified and the methodological approach was appropriate. However, the authors may wish to clarify the qualitative approach used as they mentioned thematic phenomenology in the abstract and grounded theory approach in the methods. While I am inclined to believe that their processes described fits the thematic phenomenology approach, it is possible that the use of Open code may have limited them to using grounded theory which Open code is based off. The systematic collection and analysis of data on which grounded theory is based off was not described adequately in their methods or did the authors use both approaches?

see:

Strauss, A.L. and Corbin, J.M., 1990. Basics of qualitative research: grounded theory procedures and techniques, Sage Publications, Newbury Park, CA, USA.

Strauss, A. and Corbin, J., 1994. Grounded theory methodology: An overview.

Reviewer #3: The manuscript is scientifically sound and draws conclusions relevant to the findings. It will assist policy makers in ensuring community information about child eye health practices is improved upon

Reviewer #4: The study titled “Exploring the knowledge, attitudes, and practice towards child eye health: A qualitative analysis of parent experience focus groups” was found to be of appropriate content for the journal and its audience.

The purpose of the study has been clearly stated to report on the Knowledge, attitudes and practice among parents towards Child Eye Health in Northwest Ethiopia.

The study objectives, outlined methodology and results were clearly and well reported.

However, a few observations, corrections and future recommendations are listed below.

There is need to further expatiate on the justification for this study with particular emphasis on why a qualitative approach and what this would buttress to the study.

The study also needs to highlight past studies on knowledge, attitudes and practice among parents towards CEH in Ethiopia and state what differentiates this study from others or state if there is paucity of such studies.

The methodology has been well explained, however a few concerns.

There are concerns to the fact that a purposive sampling was used to select the participants but there clearly seems to be a selection bias towards males as there were 55 males and 16 females. In the African context, females are usually seen to be the ones that take care of the home and the children at home, so its preferable that there should be a fair representation of both sexes.

Also because of the family dynamics and cultural practice in Arica, it’s advisable that females and males have separate focus group discussion, so females can be able to express themselves better as in a number of African cultures, women have been seen not to fully express their views or be open to speak freely in the presence of males.

Details of the methodology are sufficient and some recommendations above should be considered especially towards subsequent studies.

The manuscript was well written and clear enough to be accessible to non-specialist.

There is need for the authors to conform to a uniform referencing style as recommended by the journal and a few references were found not to be complete.

A few additional corrections have been made in the body of the resubmitted document.

This study design was well explained and provides an in-depth parents-based opinions with regards to child eye health and objectives of the study were clearly met.

I recommend this manuscript for publication after these minor corrections have been made and also recommend some consideration to future studies, however the sex selection bias can be stated as a limitation to the study.

6. PLOS authors have the option to publish the peer review history of their article (what does this mean?). If published, this will include your full peer review and any attached files.

Reviewer #1: No

Reviewer #2: No

Reviewer #3: **Yes: **Prof Adedayo Omobolanle Adio MD

Reviewer #4: No

---

## [Author Response · Author response to Decision Letter 0]

25 Sep 2023

22 September 2023

Editorial Board

PLOS One Journal,

RE: PONE-D-23-07767-Exploring the knowledge, attitudes, and practice towards child eye health: A qualitative analysis of parent experience focus groups

Thank you for reviewing our article: " Exploring the knowledge, attitudes, and practice towards child eye health: A qualitative analysis of parent experience focus groups". We respond to the comments in a point-by-point fashion below.

Reviewer: 1

The study looked at exploring parental KAP on children’s vision.

1.The literature review could be more informative by focusing on the topic itself – parental KAP towards CEH. There is very little on this in the introduction.

RESPONSE: We now include additional literature on parental KAP in the introduction section (Pages 2-3, lines 89-131).

2.What exactly is the novelty of this research? It seems like a lot has been known on this topic.

Alrasheed SH, Naidoo KS, Clarke-Farr PC. Childhood eye care services in South Darfur State of Sudan: Learner and parent perspectives. African Vision and Eye Health. 2016 Jan 1;75(1):1-3.

Malek Shahenaz M, Karelia Bharti N. Knowledge, Attitude And Practice On Myths And Belief Of Parents, Regarding Childhood Illness In Outdoor Unit Of Gynaecology Department Of A Tertiary Care Teaching Hospital.

Molla A, Meshesha B, Delelegn D, Reddy PS. Knowledge, attitudes and practice (KAP) and associated factors of parents/guardians on childhood eye care, Sidama Zone, Southern Ethiopia. Int J Sci Res. 2014;3:339–42.

RESPONSE: Thank you for these articles; we wish to point out that none of the suggested references directly address knowledge, attitudes and practices of CEH in Northwestern Ethiopia, which has a distinct culture and societal norms than the rest of the country. We have more clearly outlined the rationale of our study to address this (Page 4 lines 137-142)

3.To improve the state of CEH, there are plenty of factors – parental KAP is one of them. Can the authors describe these factors in their context, how they have/have not been dealt with and how parental KAP fits in the puzzle?

RESPONSE: Thanks for the suggestion. We described the factors in their context in a revised paragraph in the introduction (Page 3 lines 80-88)

4.How was the sample size of 7 focus groups determined? No sampling frame was provided.

RESPONSE: We now indicate a purposive sampling technique was used to recruit study participants (Page 5 line 174). Focus groups were terminated when data saturation was reached (Page 5, line 191-192).

5.There is no such thing as no risk in research – perhaps minimal risks?

RESPONSE: We have revised this to indicate there was minimal risk to participants in this study. (Page 5 line 202)

6.How was “Knowledge about child eye problems; attitude related themes” not fall under Attitude? A confusing way of presenting your findings. 

RESPONSE: This was a typographical error. We have revised the section (Page 6 lines 257-263)

7.It is uncommon to present findings both in table 2 and repeat them in the narrative. Choose one.

RESPONSE: Table 2 was removed from the main manuscript and placed in Supplementary Table 2.

8.The subthemes could be more informative – e.g., instead of Source of Information, it could be “Parents accessed child eye health information from X,Y,Z.

RESPONSE: Thanks for the suggestion. The subthemes were modified to be more informative (Supplementary Table 2 and Results section). 

9.I would be interested to look at the codebook seeing the dataset has not been shared.

RESPONSE: We have uploaded the codebook to the manuscript submission site; if can be included as a Supplementary File if the editorial team wishes to do so.

10.“According to a 2017 study at a blind school in the same region as the current study, corneal scarring from vitamin A deficiency and measles are the leading causes of blindness [25]. Yet these were not noted by the participants in our study, who instead identified trachoma, glaucoma, and trauma as the primary causes of childhood blindness.” – perhaps your participants do not have children from the blind schools? Please substantiate your findings with relevant literature.

RESPONSE: We have revised for clarity: “This difference warrants further investigation, since in LMIC most research on childhood blindness is performed in schools for the blind, yet the conclusions drawn may not be representative of the community, potentially limiting the generalizability of the findings.” (Page 13, lines 516-519).

11.The primary sources of information, according to study participants, were the media and health professionals. However, they all acknowledged that they lack sufficient knowledge regarding children's eye health. This may be the cause of parents misunderstanding the etiologies and common causes of child eye blindness in their setting. – How do you relate misunderstanding the etiologies to sources of information being media and health professionals?

RESPONSE: We have revised this entire section to more clearly indicate the relationship between sources of information and knowledge. (Pages 12, lines 479-499)

12.Overall, the Discussions section is very sporadic with multiple themes in one paragraph. The team could mainstream one point per paragraph.

RESPONSE: The discussion section was thoroughly revised and re-organized by 5 main action items resulting from our thematic analysis.

13.You might need to consult a native English speaker to smoothen the academic writing and grammatical errors.

RESPONSE: We have revised the language throughout the manuscript to remove grammatical errors and improve style.

Reviewer: 2

The authors set out to explore the KAP of parents regarding their ocular health seeking behaviour on behalf of their children. The premise of the study is justified and the methodological approach was appropriate. However, the authors may wish to clarify the qualitative approach used as they mentioned thematic phenomenology in the abstract and grounded theory approach in the methods. While I am inclined to believe that their processes described fits the thematic phenomenology approach, it is possible that the use of Open code may have limited them to using grounded theory which Open code is based off. The systematic collection and analysis of data on which grounded theory is based off was not described adequately in their methods or did the authors use both approaches?

see:

Strauss, A.L. and Corbin, J.M., 1990. Basics of qualitative research: grounded theory procedures and techniques, Sage Publications, Newbury Park, CA, USA.

Strauss, A. and Corbin, J., 1994. Grounded theory methodology: An overview.

RESPONSE: We used a thematic phenomenological approach for analysis; the methods section was corrected accordingly (Page 5, line 196).

Reviewer #3:

The manuscript is scientifically sound and draws conclusions relevant to the findings. It will assist policy makers in ensuring community information about child eye health practices is improved upon

RESPONSE: Thank you so much for the encouraging feedback.

Reviewer #4:

The study titled “Exploring the knowledge, attitudes, and practice towards child eye health: A qualitative analysis of parent experience focus groups” was found to be of appropriate content for the journal and its audience.

The purpose of the study has been clearly stated to report on the Knowledge, attitudes and practice among parents towards Child Eye Health in Northwest Ethiopia.

The study objectives, outlined methodology and results were clearly and well reported.

However, a few observations, corrections and future recommendations are listed below.

RESPONSE: Thank you so much for the encouraging feedback.

1. There is need to further expatiate on the justification for this study with particular emphasis on why a qualitative approach and what this would buttress to the study. 

The study also needs to highlight past studies on knowledge, attitudes and practice among parents towards CEH in Ethiopia and state what differentiates this study from others or state if there is paucity of such studies.

RESPONSE: We have elaborated on the rationale for this study (Page 4 lines 135-142) and provided more information on past studies on this topic (Pages 2-3, lines 89-131).

The methodology has been well explained, however a few concerns.

There are concerns to the fact that a purposive sampling was used to select the participants but there clearly seems to be a selection bias towards males as there were 55 males and 16 females. In the African context, females are usually seen to be the ones that take care of the home and the children at home, so its preferable that there should be a fair representation of both sexes.

Also because of the family dynamics and cultural practice in Arica, it’s advisable that females and males have separate focus group discussion, so females can be able to express themselves better as in a number of African cultures, women have been seen not to fully express their views or be open to speak freely in the presence of males.

RESPONSE: In Ethiopia, unlike in other African countries, the fathers are usually caregivers of children undergoing medical treatment, and we have seen a male preponderance in study participation in our past research as well. We do agree that this presents an undesirable gender imbalance in the study, and have discussed that in the limitations section (Page 16, lines 762-766).

Details of the methodology are sufficient and some recommendations above should be considered especially towards subsequent studies.

The manuscript was well written and clear enough to be accessible to non-specialist.

There is need for the authors to conform to a uniform referencing style as recommended by the journal and a few references were found not to be complete.

RESPONSE: We have updated the references accordingly.

A few additional corrections have been made in the body of the resubmitted document.

This study design was well explained and provides an in-depth parents-based opinions with regards to child eye health and objectives of the study were clearly met.

RESPONSE: Thank you for this helpful feedback. 

I recommend this manuscript for publication after these minor corrections have been made and also recommend some consideration to future studies, however the sex selection bias can be stated as a limitation to the study.

RESPONSE: We have indicated the gender bias as a limitation to the study.

Thank you for the opportunity to be considered for publication in your journal. We look forward to your response.

Sincerely,

Dr. Sadik T. Sherief.

---

## [Editor Report · Decision Letter 1]

17 Oct 2023

Exploring the knowledge, attitudes, and practice towards child eye health: A qualitative analysis of parent experience focus groups

PONE-D-23-07767R1

Dear Dr. SHERIEF,

We’re pleased to inform you that your manuscript has been judged scientifically suitable for publication and will be formally accepted for publication once it meets all outstanding technical requirements.

Kind regards,

Ugochukwu Anthony Eze

Academic Editor

PLOS ONE
---

## [Editor Report · Acceptance letter]

26 Oct 2023

PONE-D-23-07767R1 

Exploring the knowledge, attitudes, and practice towards child eye health: A qualitative analysis of parent experience focus groups 

Dear Dr. Sherief:

I'm pleased to inform you that your manuscript has been deemed suitable for publication in PLOS ONE. Congratulations! Your manuscript is now with our production department. 

Kind regards, 

on behalf of

Dr. Ugochukwu Anthony Eze 

Academic Editor

PLOS ONE